# Identification of the shared gene signatures and pathways between sarcopenia and type 2 diabetes mellitus

**Shiyuan Huang, Chunhua Xiang, Yi Song** *

Department of Geriatrics, Union Hospital, Tongji Medical College, Huazhong University of Science and Technology, Wuhan, China

* yisong@hust.edu.cn

## Abstract

### Background

Sarcopenia is characterized by the age-associated loss of skeletal muscle mass and strength that develops progressively and plays an important role in the disability of the elderly. It has received growing attention over the last decade and has been implicated as both a cause and consequence of type 2 diabetes mellitus (T2DM). The existence of T2DM could increase the risk of developing sarcopenia through multiple mechanisms including advanced glycation end-product accumulation. Meanwhile, sarcopenia would alter glucose disposal and may contribute to the development and progression of T2DM due to reduced muscle mass.

### Methods

We implemented transcriptomic analysis of skeletal muscle biopsy specimens in sarcopenia patients and proliferating myoblasts or differentiated myotubes from individuals with T2DM. Related microarray data were selected from Gene Expression Omnibus (GEO) to screen the genes, which were differentially expressed for sarcopenia and T2DM. Multiple combinatorial statistical methods and bioinformatics tools were used to analyze the common DEGs. Meanwhile, functional enrichment analysis was also carried out. Furthermore, we constructed the protein-protein interaction (PPI), as well as transcription factor (TF)-gene interactions network and TF-miRNA coregulatory network. Finally, based on the common DEGs, drug compounds were speculated using the Drug Signatures database (DSigDB).

### Results

A total of 1765 and 2155 DEGs of sarcopenia and T2DM were screened, respectively. 15 common genes (LXN, CIB2, PEA15, KANK2, FGD1, NMRK1, PLCB1, SEMA4G, ADARB1, UPF3A, CSTB, COL3A1, CD99, ETV3, FJX1) correlated with sarcopenia and T2DM simultaneously were then identified, and 3 genes (UPF3A, CSTB and PEA15) of them were regarded as hub genes. Functional enrichment analysis revealed several shared pathways between two diseases. In addition, according to the TF-gene interactions network and TF-

**Data Availability Statement:** Data are available from the NCBI Gene Expression Omnibus (accession numbers GSE1428 and GSE166467).

**Funding:** This work was financially supported by the National Natural Science Foundation of China

(81901429). The funders had no role in study design, data collection and analysis, decision to publish, or preparation of the manuscript.

miRNA coregulatory network, part of TF and miRNA may be identified as key regulator in sarcopenia and T2DM at the same time (e.g., CREM and miR-155). Notably, drug compounds for T2DM and sarcopenia were also suggested, such as coenzyme Q10.

## Conclusion

This study revealed that sarcopenia and T2DM may share similar pathogenesis and provided new biological targets and ideas for early diagnosis and effective treatment of sarcopenia and T2DM.

## Introduction

Primarily aging-associated skeletal muscle changes are muscle atrophy, usually accompanied by sarcopenia. Sarcopenia is characterized by the age-related progressive loss of skeletal muscle mass and strength which results in muscle weakness, restricted mobility, and increased susceptibility to injury [1]. It is estimated that about 50 million people suffering from sarcopenia in the world at present, and the number is expected to reach 500 million by the year 2050. Data show that one-third of people aged 65 and above had sarcopenia (the incidence rate was 14% to 33%), and the prevalence rate of the elderly aged 80 years and above was as high as 50%-60% [2]. The concept of 'Sarcopenia' was first defined as the age-related loss of muscle mass by Rosenberg in 1997 [3]. Later, organizations, such as the International Working Group on Sarcopenia (IWGS) and the European Working Group on Sarcopenia in Older People (EWG-SOP), expanded the definition of sarcopenia with muscle function, including muscle strength and physical performance [4,5]. However, it is hard to reach an agreement on the definition of sarcopenia, since the experts in these organizations propose different thresholds and diagnostic tests for the evaluation of muscle mass, muscle strength, and physical performance. Several reasons that contribute to sarcopenia include increased inflammation [6], excessive oxidative stress, mitochondrial dysfunction [7], reduced muscle capillarization [8], and changes in dietary intake of protein [9].

T2DM, a common metabolic disease, is usually accompanied by insulin resistance, activated inflammation, advanced glycation end-product (AGE) accumulation and increased oxidative stress. These mechanisms would result in activated inflammation, mitochondrial and vascular dysfunction, and impairments in protein metabolism, which cause a problem to muscle health, e.g., muscle mass, muscle strength, muscle quality, and muscle function [10]. Specifically, the anabolic action of insulin in skeletal muscle may be progressively lost in T2DM due to the impaired insulin sensitivity. Additionally, the impaired insulin action may induce increased protein degradation and decreased protein synthesis, which leads to reduction in muscle mass and strength [10]. Moreover, the results from the hind limb muscles of diabetic mice revealed that the decline in muscle mass, muscle endurance and regenerative capacity were related to AGEs accumulation [11]. Thus, the risk of sarcopenia is increased in T2DM patients [10,12]. On the other hand, skeletal muscle is essential for glucose clearance and is in charge of over 80% of glucose uptake from postprandial glucose load, and consequently, it has been proved that sarcopenia would alter glucose disposal by lowering muscle mass and also increase localized inflammation, which may contribute to the development and progression of T2DM [13].

Since various bidirectional associations between sarcopenia and T2DM exist, and the fact that one condition can increase the possibility of developing the other [12], our study is aiming

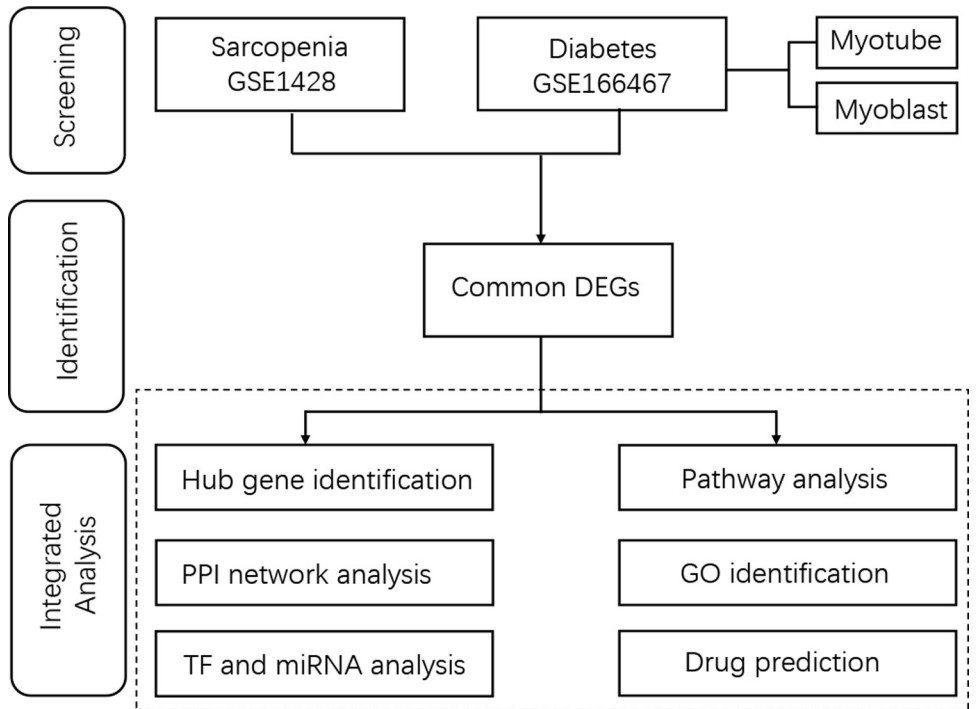

**Fig 1. Workflow of the whole study.** DEGs, differentially expressed genes; GO, Gene Ontology; PPI, protein-protein interaction; TF, transcription factor.

to find biological pathways and the relationship between sarcopenia and T2DM and provide a new idea for the diagnosis and treatment of sarcopenia. Firstly, two datasets, GSE1428 [14] and GSE166467 [15], were selected for finding DEGs for regulating skeletal muscle in sarcopenia and T2DM, respectively. Then, common DEGs were found out, and with these common DEGs, further pathway enrichment analysis was performed to investigate the biological processes of genome-based expression. The network of protein-protein interactions (PPIs) was built from these common DEGs to select the hub genes. Then, transcription factors and miRNA were also traced based on the common DEGs in the PPIs network. Finally, potential drugs were suggested via the DSigDB database. The sequential workflow of our research was presented in Fig 1.

## Material and methods

### Data collection

GSE1428 and GSE166467 datasets were assembled from the GEO database [16]. GPL96 (Affymetrix Human Genome U133A Array) platform was applied to GSE1428 dataset microarray analysis [14] and the GPL10558 (Illumina HumanHT-12 V4.0 expression bead chip) platform was used for GSE166467 [15]. GSE1428 dataset, which illustrates skeletal muscle sarcopenia in transcriptional responses, was contributed by Giresi et al. GSE166467 for T2DM was presented by Ling et al. The skeletal muscle sarcopenia dataset (GSE1428) provided microarray data from vastus lateralis muscle biopsies of young and old people. The T2DM dataset (GSE166467) contained data from both proliferating myoblasts and differentiated myotubes from individuals with T2DM and controls. In addition, before the DEGs analysis, the RMA algorithm was used to normalize the datasets from raw data if necessary.

## Identification of common DEGs between sarcopenia and T2DM

We achieved the identification of DEGs for the GSE1428 and GSE166467 datasets using GEO-query [17] and limma [18] packages in R. P-value<0.05 was considered as cutoff criteria to determine significant DEGs. Besides, common DEGs were also acquired and displayed in venn diagram with R software.

## Enrichment analysis of gene ontology and pathways

Gene set enrichment analysis is a comprehensive resource for collected gene sets that accumulate biological pathways for biological discoveries [19]. Gene ontology is a kind of functional enrichment clustered into three subsections of biological process, molecular function and cellular component [20]. In addition, four major databases, including KEGG (Kyoto Encyclopedia of Genes and Genomes), WikiPathways [21], Reactome [22] and BioCarta, were used to specify the shared pathways between sarcopenia and T2DM. GO terms and enriched signaling pathways were completed through the Enrichr web server (https://amp.pharm.mssm.edu/Enrichr/) for shared DEGs. The top 10 listed pathways with the smallest P-value are exhibited.

## PPI network construction

Finding the functions of protein is the primary step in systems biology and drug discovery [23]. Protein-protein interaction (PPI) and the obtained network are crucial in the field of biological processes at many levels of cellular structure and function, including basic metabolism and cell differentiation [24,25]. Similar DEGs in sarcopenia and T2DM were identified through the NetworkAnalyst platform (https://www.networkanalyst.ca/) and organized into a visual PPIs network [26]. Then, the PPIs network was further visualized and integrated with the Cytoscape platform (https://cytoscape.org/) to analyze the protein interactions and genetic interactions [27].

## Establishment of the topological algorithm and identification of hub nodes

In this work, the highly interconnected hub nodes [28] were determined by cytoHubba, a plugin of Cytoscape software. Nodes in the network could be ranked by 11 topological algorithms with cytoHubba (https://apps.cytoscape.org/apps/cytohubba) [29]. The prominent modules are located at the position in the PPI network where the interconnect density of the hub nodes is highest.

## Recognition of related transcription factors and miRNAs

TFs and miRNAs constitute two major regulation modes of gene expression, including transcription and post transcription. To choose the TFs and miRNAs, a co-regulatory network has been visualized by the NetworkAnalyst platform, which has been widely used as a bioinformatics tool [30,31].

## Evaluation of candidate drugs

Drug compounds recognition is the vital module of this study. According to the common DEGs for sarcopenia and T2DM which are determined in the PPIs network, drug molecules could be predicted based on the DSigDB database [32]. In this work, the access of the DSigDB database was obtained on the Enrichr platform (https://maayanlab.cloud/Enrichr/), which is widely used to represent multiple visualization details on gathered functions for the genes [33].

# Results

## Identification of DEGs and shared genes between sarcopenia and diabetes

GSE1428 dataset was normalized by the RMA algorithm (S1 Fig) before DEGs were explored for sarcopenia. A total of 1765 DEGs were obtained with 1055 upregulated and 710 downregulated genes, which satisfied the screening criteria. For T2DM, the mRNA expression level in either proliferating myoblasts or differentiated myotubes from GSE166467 with T2DM (n = 13) versus controls (n = 13) was compared. Collected 147 genes for interaction genes of sarcopenia versus proliferating myoblasts and 54 genes for interaction genes of sarcopenia versus differentiated myotubes were compared using R, followed by identification of 15 common DEGs (LXN, CIB2, PEA15, KANK2, FGD1, NMRK1, PLCB1, SEMA4G, ADARB1, UPF3A, CSTB, COL3A1, CD99, ETV3 and FJX1) (S1 Table). Venn diagram displayed the overlap of DEGs from sarcopenia, differentiated myotubes and proliferating myoblasts (Fig 2A). The heat map for the shared common genes showed paralleled transcriptional signature among most of these genes according to the log fold change (Fig 2B).

## Functional enrichment analysis

The analysis of gene functional enrichment was carried out on the Enrichr platform. The ongoing study analyzed GO terms and enriched pathways for 15 common DEGs (LXN, CIB2, PEA15, KANK2, FGD1, NMRK1, PLCB1, SEMA4G, ADARB1, UPF3A, CSTB, COL3A1, CD99, ETV3, FJX1). The data justified that the common DEGs are mostly enhanced in the regulation of G1/S transition of mitotic cell cycle and Rho protein signal transduction for the biological process subsection. In the molecular function group, the common DEGs are mainly enriched in endopeptidase inhibitor activity and protease binding factors. Cellular component study exhibited that common DEGs located significantly in the nucleolus. Analysis result from KEGG, WikiPathway, Reactome and BioCarta pathway was also obtained. The information attained showed the AGE-RAGE signaling pathway in T2DM complications significantly (P<0.01) assembled in the KEGG pathway database. Functional analysis results of GO

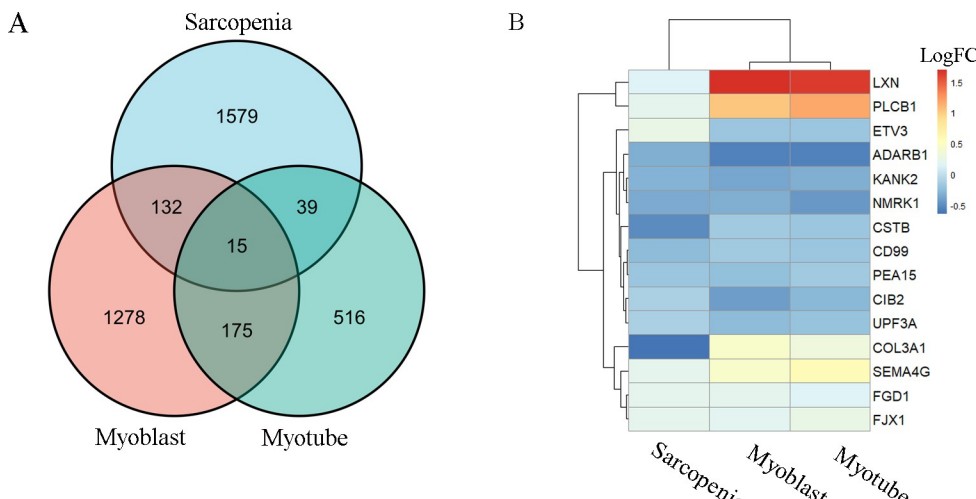

**Fig 2. Identification of shared DEGs between sarcopenia and diabetes.** (A) Overlap of DEGs represented through a Venn diagram. 15 genes were found common from 147 genes for interaction genes of sarcopenia versus proliferating myoblasts and 54 genes for interaction genes of sarcopenia versus differentiated myotubes. (B) Heat map for the shared DEGs of sarcopenia and T2DM according to the log-fold change.

(Fig 3A) and pathway (Fig 3B) were displayed according to -log10(P value) (detailed in S2–S6 Tables).

## PPI network to identify hub genes

The InnateDB database on NetworkAnalyst was employed to construct PPI network of the common DEGs, which was then imported into Cytoscape software for visualization and optimization. The PPIs network contained 126 nodes and 127 edges (Fig 4), and it was constructed for hub gene identification and drug molecules detection for sarcopenia and diabetes in the following steps.

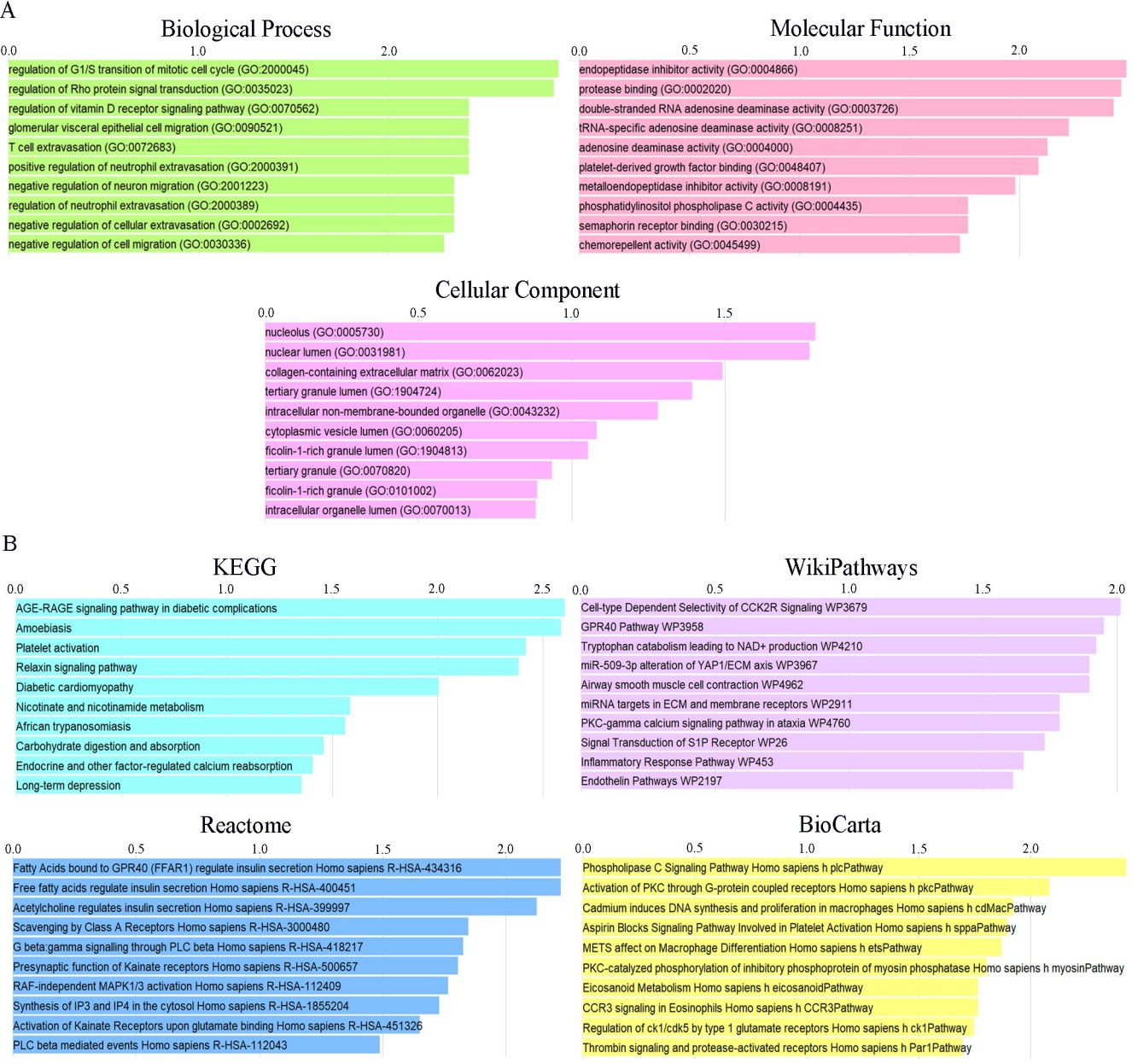

**Fig 3. Functional enrichment analysis of common genes.** (A) Enriched GO terms. (B) Pathway analysis through databases of KEGG, WikiPathway, Reactome and BioCarta. The results were displayed according to -log10(P value).

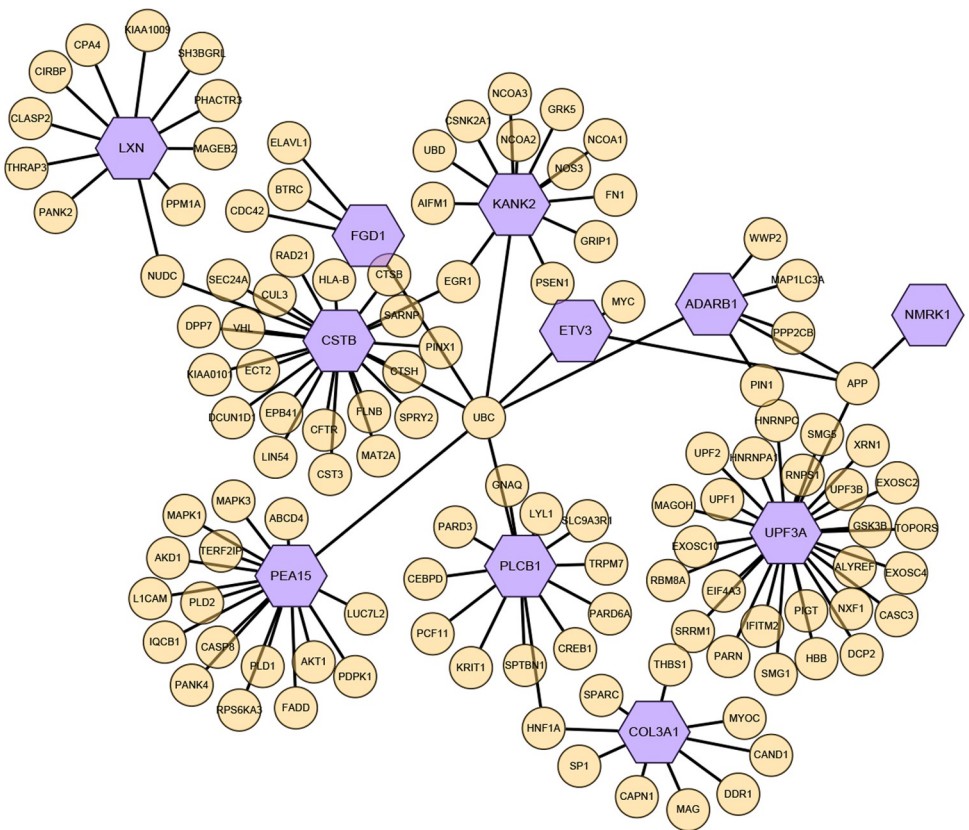

**Fig 4. PPI network for common DEGs shared by two diseases.** Nodes in purple color indicated common DEGs.

## Identification of hub genes

Hub genes were screened regarding their degree value from the PPI network in cytohubba, which was a plugin of Cytoscape software. The top three identified hub genes are UPF3A, CSTB and PEA15, which were all affiliated to common genes of sarcopenia and proliferating myoblasts of T2DM. The network consisted of 72 nodes and 73 edges. Interactions between interrelated proteins (Fig 5) and the topological analysis result were presented (S7 Table).

## TF-gene interaction network

TF-gene interaction network was generated using the NetworkAnalyst web tool. For the 15 common DEGs (LXN, CIB2, PEA15, KANK2, FGD1, NMRK1, PLCB1, SEMA4G, ADARB1, UPF3A, CSTB, COL3A1, CD99, ETV3, FJX1), the TF-genes were identified (Fig 6). The network contained 161 nodes and 230 edges. CSTB, SEMA4G and ADARB1 were regulated by 62, 38 and 31 TF-genes, respectively.

## TF-miRNA coregulatory network

The main TF-miRNA coregulatory network based on the 15 common DEGs was produced via NetworkAnalyst. The network created comprised 289 nodes and 409 edges including 15 common DEGs (S2 Fig). Details were displayed in S8 Table. This interaction can be used to analyze the regulation factor for sarcopenia combined with T2DM. Moreover, the subnetwork, which specifically include only hub genes (UPF3A, CSTB and PEA15) and their associated miRNA and TF (Fig 7), was extracted from the main TF-miRNA coregulatory network.

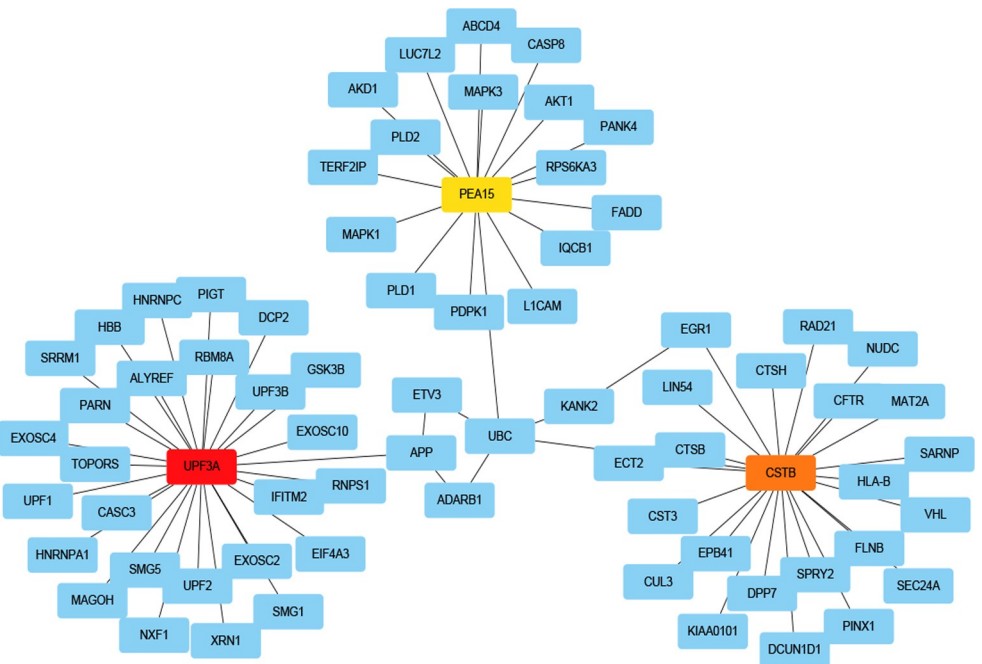

**Fig 5. Identification of hub genes from PPI network.**

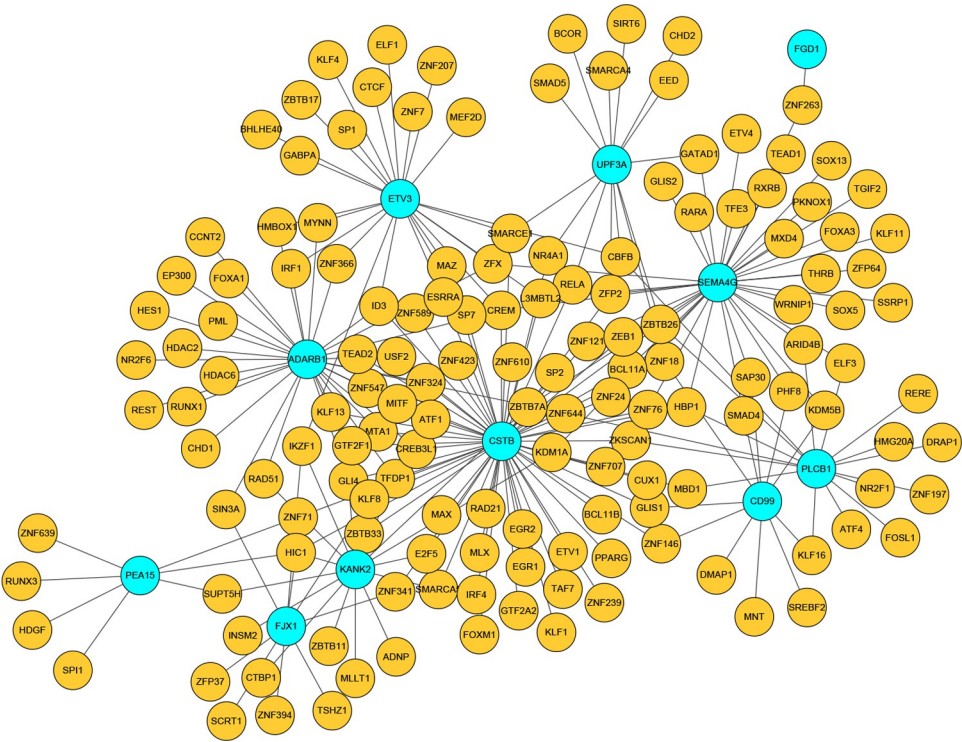

**Fig 6. TF-gene interaction network based on common DEGs.** Nodes in green color indicated common DEGs, and nodes in yellow color indicated TF genes.

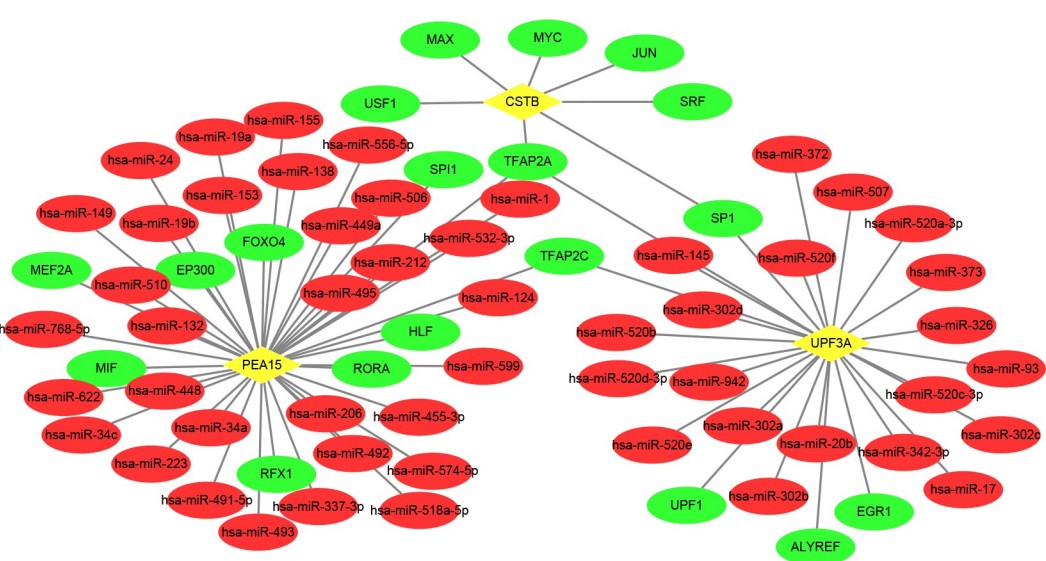

**Fig 7. TF-miRNA coregulatory network including hub genes.** The nodes with yellow diamonds are hub genes, red round nodes represent miRNA and green round nodes indicate TF.

### Identification of candidate drugs

Drug molecules were detected for 15 common DEGs from the DSigDB database on Enrichr platform. Results from the candidate drugs were generated, and these drugs represented possible common drugs for sarcopenia and diabetes. The analysis depicted that Coenzyme Q10 CTD 00001167 and dihydroergocristine HL60 UP were the two drug molecules with the highest combined score (Table 1).

### Discussion

Sarcopenia-related muscle dysfunction has a great impact on the life quality of elderly people. Changes in skeletal muscle development, homeostasis and metabolism during aging are reflected in gene expression regulation [34]. Sarcopenia has been discovered as a new diabetes complication among the elderly population, which makes it a major public health event [10]. The study was implemented to analyze microarray data of sarcopenia and T2DM. 15 common DEGs were identified (including LXN, CIB2, PEA15, KANK2, FGD1, NMRK1, PLCB1,

**Table 1. Suggested top drug compounds for sarcopenia.**

| Term | P-value | Adjusted P-value | Combined Score | Genes |
|---|---|---|---|---|
| Coenzyme Q10 CTD 00001167 | 0.008965 | 0.141336 | 611.4626003 | ETV3 |
| dihydroergocristine HL60 UP | 0.001684 | 0.085573 | 244.4667349 | LXN; PLCB1 |
| (-)-isoprenaline HL60 UP | 3.11E-04 | 0.031049 | 223.3772771 | LXN; ADARB1; PLCB1 |
| Prestwick-983 HL60 UP | 3.48E-04 | 0.031049 | 211.9492283 | LXN; ADARB1; PLCB1 |
| colforsin MCF7 DOWN | 0.022273 | 0.199012 | 186.9959577 | KANK2 |
| etacrynic acid HL60 UP | 0.023741 | 0.205246 | 171.9789874 | LXN |
| 2,2',5,5'-TETRACHLOROBIPHENYL CTD 00000481 | 0.027403 | 0.21128 | 142.3784815 | PLCB1 |
| OZONE CTD 00006460 | 0.028134 | 0.21128 | 137.5099228 | COL3A1 |
| TERT-BUTYL HYDROPEROXIDE CTD 00007349 | 2.39E-05 | 0.006409 | 130.1775809 | KANK2;COL3A1; NMRK1; UPF3A; ETV3; FJX1; CD99 |
| spironolactone CTD 00006774 | 0.029594 | 0.21128 | 128.5967412 | COL3A1 |

SEMA4G, ADARB1, UPF3A, CSTB, COL3A1, CD99, ETV3 and FJX1) to establish PPI network, as well as TF-gene interactions network and TF-miRNA coregulatory network. In addition, predicted drug compounds of sarcopenia and T2DM were suggested.

Identified 15 common DEGs were selected for exploring GO terms. GO terms were selected according to the P-values. For biological process, regulation of G1/S transition of the mitotic cell cycle, regulation of Rho protein signal transduction, regulation of vitamin D receptor signaling pathway were among the top GO term. Evidence revealed that the G1-to-S-phase transition is critical for cell proliferation [35]. Rho is also called Rho GTPases, which plays an important role in the regulation of cytoskeletal recombination and takes part in many physiological activities including cell migration, adhesion, cytokinesis, proliferation, differentiation and apoptosis [36]. Vitamin D receptor (VDR), which might be expressed in muscle fibers through vitamin D signaling, acts a crucial role in regulating myoblast proliferation, differentiation and moderate sarcopenia [37,38]. GO terms in terms of molecular function endopeptidase inhibitor activity, protease binding and double-stranded RNA adenosine deaminase activity were considered to be at the top of the list. Top GO terms based on the cellular component were nucleolus, nuclear lumen, collagen-containing extracellular matrix.

The determination of the KEGG pathway was acquired from the 15 common DEGs to find similar pathways for both sarcopenia and T2DM. Top 10 KEGG pathways included AGE-RAGE signaling pathway in diabetic complications, Amoebiasis, Platelet activation, Relaxin signaling pathway, Diabetic cardiomyopathy, Nicotinate and nicotinamide metabolism, African trypanosomiasis, Carbohydrate digestion and absorption, Endocrine and other factor-regulated calcium reabsorption, Long-term depression. The AGE-RAGE axis enhances the generation of ROS and a good deal of cytokines and chemokines, which lead to local tissue insulin resistance [39]. Therefore, this signaling pathway acts a crucial role in the pathogenesis of diabetic complications [40]. AGE, known as an aging product, also leads to sarcopenia through increasing reactive oxygen species generation [39]. At the same time, data from WikiPathways showed that the most interacted gene pathways were Cell-type Dependent Selectivity of CCK2R Signaling WP3679, GPR40 Pathway WP3958 and Tryptophan catabolism leading to NAD+ production WP4210. Results from the Reactome pathway produced Fatty Acids bound to GPR40 (FFAR1) regulate insulin secretion Homo sapiens R-HAS-434316 and Free fatty acids regulate insulin secretion Homo sapiens R-HAS-400451. BioCarta database hinted that these common genes were enriched in Phospholipase C Signaling Pathway Homo sapiens h plcPathway.

PPI network analysis was the vital section, which helped us detect potential hub genes involved in the shared mechanisms of sarcopenia and T2DM. Analysis for PPI also came into being for 15 common DEGs and results revealed that owing to possessing a high interaction rate and high degree value, UPF3A, CSTB and PEA15 genes were selected as hub genes, respectively. PEA15 encodes a death effector domain-containing protein, which acts as a negative regulator for apoptosis. This encoded protein may contribute to insulin resistance in glucose uptake, since it is an endogenous substrate for protein kinase C and overexpressed in T2DM [41].

TFs are proteins that control the transcription of DNA into RNA by attaching to a particular DNA sequence, hence it is essential for regulatory biomolecules [42]. Furthermore, miRNAs participate in the regulation of protein expression mainly through binding to target sites on an mRNA transcript and inhibiting its translation [43]. TF-genes and miRNAs play key roles in the ratio of transcription and RNA silencing on the post-transcription, respectively. Thus, they are significant regulatory biomolecules and even potential biological markers [44]. TF-gene interaction network was acquired from the common DEGs. According to the network, CSTB has a high interaction rate with other TF-genes, displaying a degree value of 62 in

the network. Among the regulators, CREM had significant interaction with degree value of 4. Mechanistically, oxidative stress activation in T2DM enhanced CREM expression [45]. In experimental mouse models, CREM expression was associated with preventing inflammation and myofibrillar protein degradation that affect sarcopenia [46]. Besides, TF-miRNA coregulatory network analysis indicated relationships among the common DEGs, TFs and miRNAs through visualized TF-miRNA coregulatory network. As a result, 206 miRNAs and 68 TF-genes were revealed in this study. Among the most interacted TFs, EGR1 has a high degree value of 4. The expression of EGR1 is rapidly changed by mechanical stimulation. The induction of Sirt1 by EGR1 is necessary for moving away from the superfluous reactive oxygen species, which is produced by the mechanical stimulus. This activation is lost in aged animals due to the loss of EGR1. Therefore, the decreased expression of EGR1 in the elderly may contribute to reduced muscle function in sarcopenia [47]. The change of miRNA expression in T2DM patients has been established by many researchers, and their results revealed that miR-155 is crucial for insulin sensitivity regulation of liver, adipose tissue, and skeletal muscle. Moreover, a series of clinical researches have also found low miR-155 levels in the serum of patients with T2DM, which may lead to insulin resistance [48]. Meanwhile, increasing evidence has revealed that miRNAs are differentially expressed in sarcopenia of the elderly, plasma miR-155 level in the sarcopenia group was significantly reduced compared to the non-sarcopenia group [49].

Based on the DSigDB database, drug compounds were put forward. Among all the predicted drugs, our study highlighted the top 10 significant drugs. Coenzyme Q10 CTD 00001167, dihydroergocristine HL60 UP, (-)-isoprenaline HL60 UP, Prestwick-983 HL60 UP, colforsin MCF7 DOWN, etacrynic acid HL60 UP, 2,2',5,5'-TETRACHLOROBIPHENYL CTD 00000481, OZONE CTD 00006460, TERT-BUTYL HYDROPEROXIDE CTD 00007349 and spironolactone CTD 00006774 were the peak drug candidates for sarcopenia and T2DM. A series of studies have shown that coenzyme Q10 (CoQ10) could prevent oxidative damage, optimize mitochondrial functionality and anti-inflammatory effect [50]. These mechanisms are related to the physiology and biochemistry of aging muscle and also the potential function in preventing sarcopenia [51]. The previous study has proven that combining CoQ10 with metformin can improve glycemic control by reducing oxidative stress and improving the mitochondria morphology [52]. Furthermore, it was found in the mouse skeletal muscle transplants that isoprenaline increased the volume of regenerated muscle due to the hypertrophic effect [53]. Insulin-stimulated glucose transport activity in rat adipocytes was inhibited by isoprenaline [54]. Therefore, isoproterenol may alleviate sarcopenia by improving muscle metabolism and muscle function.

The current study used a lot of bioinformatics methodologies with GSE1428, which compared biopsy samples of skeletal muscle between the elderly and young people, and GSE166467 which indicated mRNA expression in either proliferating myoblasts or differentiated myotubes from individuals with T2DM. We have implemented DEGs analysis between two databases and tried to find the mechanisms by which two diseases interact. Indeed, different reasons (e.g., impaired insulin sensitivity, AGEs, subclinical inflammation and mitochondrial dysfunction) may be related to this association. Therefore, this work is expected to provide guidance for healthcare professionals to diagnose sarcopenia at an early stage for elderly patients with T2DM. Furthermore, the drug targets are suggested through the identification of hub genes. For example, CoQ10 has been shown to be effective in the animal models of both T2DM and sarcopenia treatment [50,51]. Biguanides, a first-line hypoglycemic agent, may protect the development of sarcopenia, which is revealed by observational cross-sectional research [55]. It is expected that more mature T2DM treatment may inspire the treatment of sarcopenia and these drugs should be considered for further verification by clinical trials. In

the future, it is hoped that specific drugs will be developed to treat sarcopenia in addition to protein supplementation and exercise.

## Conclusion

This is the first study to explore the relationship between sarcopenia and T2DM using transcriptome analysis. We have recognized the common genes between sarcopenia and T2DM to explore the association of these two diseases. Analysis revealed that sarcopenia and T2DM have similar pathogenesis including increased inflammation, excessive oxidative stress, mitochondrial dysfunction. Moreover, the existence of one disease may increase the risk of developing the other. Besides, some drug targets, which are identified based on the hub genes, are logically selected since they are probably candidate for drugs that already sanctioned. In conclusion, our research provides new biological targets and ideas for early diagnosis and effective treatment of sarcopenia combined with T2DM.

## Supporting information

**S1 Fig. Normalization of the GSE1428 microarray data.**
(TIF)

**S2 Fig. The main TF-miRNA coregulatory network.** The nodes with yellow diamonds are common DEGs, red round nodes represent miRNA and green round nodes indicate TF.
(TIF)

**S1 Table. Differentially expressed genes.**
(XLSX)

**S2 Table. GO annotation analysis details.**
(XLSX)

**S3 Table. KEGG enrichment analysis details.**
(XLSX)

**S4 Table. WikiPathway enrichment analysis details.**
(XLSX)

**S5 Table. Reactome enrichment analysis details.**
(XLSX)

**S6 Table. BioCarta enrichment analysis details.**
(XLSX)

**S7 Table. Topological results for top three hub genes.**
(XLSX)

**S8 Table. Details of the main TF-miRNA coregulatory network.**
(XLSX)

## Acknowledgments

We thank all GEO data builders and data contributors, as well as the team that built the Enrichr online analysis page.

## Author Contributions

**Conceptualization:** Yi Song.

**Data curation:** Shiyuan Huang, Yi Song.

**Writing – original draft:** Shiyuan Huang, Yi Song.

**Writing – review & editing:** Chunhua Xiang, Yi Song.

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
