## [Decision Letter · Decision Letter 0]

17 Jan 2022

PONE-D-21-34818Identification of the shared gene signatures and pathways between sarcopenia and type 2 diabetes mellitusPLOS ONE

Dear Dr. Song,

Thank you for submitting your manuscript to PLOS ONE. After careful consideration, we feel that it has merit but does not fully meet PLOS ONE’s publication criteria as it currently stands. Therefore, we invite you to submit a revised version of the manuscript that addresses the points raised during the review process.

We look forward to receiving your revised manuscript.

Kind regards,

Kanhaiya Singh, Ph.D

Academic Editor

PLOS ONE

Journal Requirements:

 “The National Natural Science Foundation of China (81901429).“

4. We note that Figure 1 in your submission contain copyrighted images. All PLOS content is published under the Creative Commons Attribution License (CC BY 4.0), which means that the manuscript, images, and Supporting Information files will be freely available online, and any third party is permitted to access, download, copy, distribute, and use these materials in any way, even commercially, with proper attribution. For more information, see our copyright guidelines: http://journals.plos.org/plosone/s/licenses-and-copyright.

Additional Editor Comments:

Although the Reviewers have found this study interesting, they have recommended some modifications.

Reviewers' comments:

Reviewer's Responses to Questions

**Comments to the Author**

1. Is the manuscript technically sound, and do the data support the conclusions?

Reviewer #1: Yes

Reviewer #2: Yes

2. Has the statistical analysis been performed appropriately and rigorously? 

Reviewer #1: Yes

Reviewer #2: Yes

3. Have the authors made all data underlying the findings in their manuscript fully available?

Reviewer #1: Yes

Reviewer #2: Yes

4. Is the manuscript presented in an intelligible fashion and written in standard English?

Reviewer #1: Yes

Reviewer #2: Yes

5. Review Comments to the Author

Reviewer #1: While going through the manuscript. I found it very impressive. But it has some lacking which need to be addressed.

First, the name of those miRNA which are regulating the selected hub genes are missing, network is so condensed that it is very difficult to figure out the name of those regulatory miRNA. It would be better if name of miRNA regulating these hub genes will be incorporated in result as well as discussion part.

Second, It would be much better if a subnetwork will be extracted from the main network of TF-miRNA-mRNA which specifically include only hub genes and its associated miRNA and TF. That would increase the clarity of network due to reduced nodes number and also will be more informative. Main network can be considered as supplementary data.

Third, Why the author has decided to go with only one dataset in each case?

Reviewer #2: In this manuscript the authors are looking into sarcopenia and Type 2 Diabetes Mellitus (T2DM) at the transcriptomic level. They have compared DEGs between the two conditions and identified common pathways. It has also been indicated that one condition can increase the risk of developing the other condition. The authors have also identified drugs that can be used as a potential treatment for both sarcopenia and T2DM.

Overall, the findings of this manuscript are well-supported by the data and the methods used are appropriate. The manuscript is also well-drafted. There are a few points to consider that I have outlined below:

1. Some terms and concepts have been explained in the materials and methods section. It will be better if they are explained either in the introduction or the results section.

2. Lines 139, ‘and its inhibiting translation’, please rephrase. It is confusing.

3. Line 153 should be ‘were obtained’.

4. Line 218, Table 1, please use borders for all the cells. The information currently looks haphazard.

5. Line 332, it is confusing. Please rephrase or clarify.

6. Line 354, should be ‘Acknowledgements’.

6. PLOS authors have the option to publish the peer review history of their article (what does this mean?). If published, this will include your full peer review and any attached files.

Reviewer #1: No

Reviewer #2: No

---

## [Author Response · Author response to Decision Letter 0]

7 Feb 2022

Editor

Comment: Please ensure that your manuscript meets PLOS ONE's style requirements, including those for file naming.

Respond: Corrections have been made.

Comment: Thank you for stating the following financial disclosure: “The National Natural Science Foundation of China (81901429).” Please state what role the funders took in the study. If the funders had no role, please state: "The funders had no role in study design, data collection and analysis, decision to publish, or preparation of the manuscript.” If this statement is not correct you must amend it as needed. Please include this amended Role of Funder statement in your cover letter; we will change the online submission form on your behalf.

Respond: Corrections have been made in the cover letter.

Comment: In your Data Availability statement, you have not specified where the minimal data set underlying the results described in your manuscript can be found. PLOS defines a study's minimal data set as the underlying data used to reach the conclusions drawn in the manuscript and any additional data required to replicate the reported study findings in their entirety. We will update your Data Availability statement to reflect the information you provide in your cover letter.

Respond: Corrections have been made in the cover letter.

Comment: We note that Figure 1 in your submission contain copyrighted images.

Respond: We have already changed Figure 1 to avoid the problem.

Comment: Please review your reference list to ensure that it is complete and correct. If you have cited papers that have been retracted, please include the rationale for doing so in the manuscript text, or remove these references and replace them with relevant current references. Any changes to the reference list should be mentioned in the rebuttal letter that accompanies your revised manuscript. 

Respond: Thanks, we have double checked the reference list. Reference [30] and [31] has been placed after original reference [43] in revised manuscript.

Reviewer 1

Comment: First, the name of those miRNA which are regulating the selected hub genes are missing, network is so condensed that it is very difficult to figure out the name of those regulatory miRNA. It would be better if name of miRNA regulating these hub genes will be incorporated in result as well as discussion part.

Second, It would be much better if a subnetwork will be extracted from the main network of TF-miRNA-mRNA which specifically include only hub genes and its associated miRNA and TF. That would increase the clarity of network due to reduced nodes number and also will be more informative. Main network can be considered as supplementary data.

Respond: The number of miRNA which regulated the selected hub genes was large, so we have displayed them in S10 table. Since the network of main TF-miRNA coregulatory network is so condensed, we have exhibited it as S9 Fig and extracted a subnetwork involving only hub genes and its associated miRNA and TF as Fig 7.

Comment: Third, Why the author has decided to go with only one dataset in each case?

Respond: Of course, combining information from multiple existing studies can increase the reliability and generalizability of results. However, concerns also exist at the same time. Heterogeneities caused by demographic, clinical and technical variables often exist across studies from different platform. Failure to consider these variables in the statistical models and meta-analysis can result in reduced statistical power or false positives. In this study, we chose only GSE1428 for sarcopenia and GSE166467 for T2DM to get preliminary result. We should also notice the shortcoming of the study. Firstly, analytical methods may limit the predictive capability of the result and new proof from future studies may renew current results. Secondly, our analyses are based on public datasets, and further researches about the detailed molecular mechanism are necessary. 

doi: 10.1371/journal.pmed.0050184

doi: 10.1093/nar/gkr1265

Reviewer 2

Comment: Some terms and concepts have been explained in the materials and methods section. It will be better if they are explained either in the introduction or the results section.

Respond: Corrections have been made. TFs and miRNA are explained in the discussion part in revised manuscript.

Comment: Lines 139, ‘and its inhibiting translation’, please rephrase. It is confusing.

Respond: Corrections have been made.

Comment: Line 153 should be ‘were obtained’.

Respond: Corrections have been made.

Comment: Line 218, Table 1, please use borders for all the cells. The information currently looks haphazard.

Respond: Corrections have been made.

Comment: Line 332, it is confusing. Please rephrase or clarify.

Respond: Corrections have been made in revised manuscript.

Comment: Line 354, should be ‘Acknowledgements’.

Respond: In standard American English, “acknowledgments” is spelled without the additional “e.” In British English, and most places outside of North America, “acknowledgements” is spelled with the extra “e.” After we searched some papers published on PLOS ONE recently, we found that ‘Acknowledgments’ without “e” was mainly used.

doi: 10.1371/journal.pone.0263301

doi: 10.1371/journal.pone.0263344

doi: 10.1371/journal.pone.0263307

---

## [Decision Letter · Decision Letter 1]

28 Feb 2022

Identification of the shared gene signatures and pathways between sarcopenia and type 2 diabetes mellitus

PONE-D-21-34818R1

Dear Dr. Song,

We’re pleased to inform you that your manuscript has been judged scientifically suitable for publication and will be formally accepted for publication once it meets all outstanding technical requirements.

Kind regards,

Kanhaiya Singh, Ph.D

Academic Editor

PLOS ONE

Additional Editor Comments (optional):

Reviewers' comments:

Reviewer's Responses to Questions

**Comments to the Author**

1. If the authors have adequately addressed your comments raised in a previous round of review and you feel that this manuscript is now acceptable for publication, you may indicate that here to bypass the “Comments to the Author” section, enter your conflict of interest statement in the “Confidential to Editor” section, and submit your "Accept" recommendation.

Reviewer #1: All comments have been addressed

Reviewer #2: All comments have been addressed

2. Is the manuscript technically sound, and do the data support the conclusions?

Reviewer #1: Yes

Reviewer #2: Yes

3. Has the statistical analysis been performed appropriately and rigorously? 

Reviewer #1: Yes

Reviewer #2: Yes

4. Have the authors made all data underlying the findings in their manuscript fully available?

Reviewer #1: Yes

Reviewer #2: Yes

5. Is the manuscript presented in an intelligible fashion and written in standard English?

Reviewer #1: Yes

Reviewer #2: Yes

6. Review Comments to the Author

Reviewer #1: Manuscript has been revised well and all the points have been addressed wisely. The manuscript can be considered for the publication

Reviewer #2: (No Response)

7. PLOS authors have the option to publish the peer review history of their article (what does this mean?). If published, this will include your full peer review and any attached files.

Reviewer #1: No

Reviewer #2: No

---

## [Editor Report · Acceptance letter]

2 Mar 2022

PONE-D-21-34818R1 

Identification of the shared gene signatures and pathways between sarcopenia and type 2 diabetes mellitus 

Dear Dr. Song:

I'm pleased to inform you that your manuscript has been deemed suitable for publication in PLOS ONE. Congratulations! Your manuscript is now with our production department. 

Kind regards, 

on behalf of

Dr. Kanhaiya Singh 

Academic Editor

PLOS ONE